# Association of intra-articular injection and knee arthroscopy prior to primary knee replacement with the timing and outcomes of surgery: Retrospective cohort study using data from the Clinical Practice Research Datalink GOLD database

Matthew Strang[1], John Broomfield[2], Michael Whitehouse[1,3], Setor Kunutsor[1,4], Sion Glyn-Jones[2], Antonella Delmestri[2], Ashley Blom[1,5], Andrew Judge[1,2,3]*

1 Musculoskeletal Research Unit, Translational Health Sciences, Bristol Medical School, Southmead Hospital, University of Bristol, Bristol, United Kingdom, 2 Nuffield Department of Orthopaedics, Rheumatology and Musculoskeletal Sciences, Nuffield Orthopaedic Centre, University of Oxford, Oxford, United Kingdom, 3 National Institute for Health Research Bristol Biomedical Research Centre, University Hospitals Bristol and Weston NHS Foundation Trust and University of Bristol, Bristol, United Kingdom, 4 Diabetes Research Centre, Leicester General Hospital, University of Leicester, Leicester, United Kingdom, 5 Faculty of Medicine, Dentistry & Health, University of Sheffield, Sheffield, United Kingdom

* Andrew.judge@bristol.ac.uk

## Abstract

### Background

Patients with symptomatic knee osteoarthritis may undergo non-surgical interventions such as intra-articular steroid injections and knee arthroscopy. This study aimed to investigate their association with the timing and outcomes of subsequent primary knee replacement.

### Methods and findings

Observational retrospective analysis of linked Clinical Practice Research Datalink, Hospital Episode Statistics, Patient Reported Outcome Measures (CPRD GOLD-HES-PROMS) data of 38,494 patients undergoing primary knee replacements in England. Prior use of intra-articular steroid injections and knee arthroscopy were identified. Hazard ratios (HRs) with 95% CIs were estimated for primary outcomes of revision and reoperation using Cox regression. Secondary outcomes included time from first diagnosis of ipsilateral knee osteoarthritis to knee replacement, 6-month post-operative Oxford Knee Scores (OKS), mortality (90-days and 3-months), and post-operative surgical site infection (SSI) (3-months) using linear and logistic regression. Prior steroid injections were associated with an increased risk of revision (HR = 1.25 95%CI (1.06 to 1.49)), re-operation (HR = 1.18 95%CI (1.05 to 1.32)), and SSI (HR = 3.10 95%CI (1.14 to 8.46). Timing from diagnosis of knee osteoarthritis to knee replacement was 6 months longer in patients receiving steroid injections. Knee arthroscopy was associated with an increased risk of revision (HR = 3.14 95%CI (2.64 to

**Data Availability Statement:** Electronic health records are, by definition, considered sensitive data in the UK by the Data Protection Act and cannot be shared via public deposition because of information governance restriction in place to protect patient confidentiality. Access to data is available once approval has been obtained through the individual constituent entities controlling access to the data. The primary care data can be requested via application to the Clinical Practice Research Datalink, secondary care data can be requested via application to the Hospital Episode Statistics from the UK Health and Social Care Information Centre, and mortality data are available by application to the UK Office for National Statistics. Alternatively, access to secondary care data and mortality data is available through linkage requested upon application for primary care data to the Clinical Practice Research Datalink. Information regarding linkage is available from: https://www.cprd.com/linked-data (https://www.cprd.com/research-applications).

**Funding:** The author(s) received no specific funding for this work.

**Competing interests:** All authors have completed the Unified Competing Interest form at www.icmje.org/coi_disclosure.pdf and declare: AJ reports personal fees from Freshfields Bruckhaus Derringer, personal fees from Anthera Pharmaceuticals Ltd, outside the submitted work; All other authors declare no conflicts of interest. This does not alter our adherence to PLOS ONE policies on sharing data and materials.

3.73)), re-operation (HR = 3.25 95%CI (2.89 to 3.66)), lower post-operative OKS -1.63 95% CI (-2.31 to -0.95). Both interventions were associated with a lower risk of mortality.

## Conclusions

Steroid injection and knee arthroscopy prior to primary knee replacement are each associated with worse outcomes. The observed association of lower mortality risk is suggestive of confounding by indication. The observed associations in this study could be used to inform shared decision making with patients on the treatment pathway for knee osteoarthritis.

## Introduction

Knee replacements are common and cost-effective surgical procedures for alleviating pain and the disability associated with advanced joint disease such as osteoarthritis (OA) [1]. Over 100,000 primary knee replacements are performed annually in England and Wales [2]. There has been an observed trend toward younger age at time of primary knee replacement [3]. In the United States, there is a predicted increase in primary knee replacements of 401% by 2040 [4]. In England and Wales, it is projected that primary knee replacements will increase by 117% between 2012 and 2030 [5]. Management options for OA include core (non-surgical) treatment (weight loss, a variety of interventions led by physical therapists, exercise, and oral or topical pain medications such as non-steroidal anti-inflammatory drugs), and invasive treatments such as intra-articular corticosteroid and other injections, knee arthroscopy (KA), and knee replacement or osteotomy [6]. The preferred combination or sequence of these options is not clear and may vary between patients. Patients who eventually undergo primary knee replacements may have received prior intra-articular steroid injections (IASI) or undergone prior KA to manage symptoms and potentially delay knee replacement surgery. The National Institute for Health and Care Excellence (NICE) guidance recommends the use of IASI as an adjunct to core treatments for the relief of moderate-to-severe, uncontrolled pain in people with OA [6]; around 30% of patients will have received IASI prior to knee replacement [7]. Though a number of studies have reported on the associations of prior IASI and outcomes following knee replacement, the majority of these studies were based on matched cohorts with small sample sizes that mostly evaluated infection outcomes and reported inconsistent findings [8–11]. Intra-articular injection of platelet rich plasma (PRP) has become increasingly utilised as a minimally invasive treatment option for symptomatic knee OA [12]. Previous literature has shown favourable pain and functional outcomes compared to saline, Hyaluronic Acid (HA) and IASI [13, 14]. However, more recent high quality RCTs have shown no benefit in the use of PRP when compared to placebo injections [15, 16]. NICE have published recommendations on PRP injections that do not routinely recommend its use in treating OA, due to inconsistent and limited quality evidence on efficacy [17].

There is emerging research into the potential use of injectable biomaterials in treating OA including; hydrogels, non-hydrogel polymers, and inorganic nanomaterials [18, 19]. These compounds act as highly effective carriers of drugs and bioactive factors that aim to regenerate damaged articular cartilage, with recent studies showing promising results [20].

Despite mounting evidence against the effectiveness of KA in degenerative knee pathologies and strong recommendations against its use in nearly all patients with degenerative knee disease [21], it has been reported that approximately 29% of patients receive arthroscopic knee surgery prior to primary knee replacement [22]. The current literature on the outcomes of

primary knee replacement following prior KA is inconsistent; some studies have reported adverse associations with outcomes such as revision, reoperation, and infection following subsequent knee replacement [23–25], whereas others have reported no evidence of these associations [26, 27]. Furthermore, most of these studies were limited by small sample sizes and short follow-up periods.

IASI and KA remain commonly utilised orthopaedic interventions, yet their impact on timing and outcomes following primary knee replacement is uncertain. Using a large retrospective cohort study and a comprehensive list of relevant clinical outcomes, this study aimed to investigate the association between prior IASI and KA with timing and outcomes following primary knee replacement.

## Methods

### Study design and data sources

This was an observational retrospective analysis of routinely collected linked data from the Clinical Practice Research Datalink (CPRD) GOLD database linked to English National Health Service (NHS) Hospital Episodes Statistics (HES) Admitted Patient Care (APC) and Patient Reported Outcome Measures (PROMS).

### Participants and interventions

A total of 64,071 patients undergoing primary knee replacement (total and unicompartmental) between 1st Jan 1995 and 31st December 2016 were identified in the CPRD GOLD database. We excluded patients with underweight body mass index (BMI), of less than 18.5, as there were too few patients in this category to fit the regression models in our analysis. We only included patients with linked HES-CPRD GOLD data leaving a total of 38,494 patients for the analysis of demographics, revision, re-operation, and surgical site infection. Timing intervals between diagnosis of OA and primary knee replacement was available for 14,972 patients. Linked data for 6-month pre and post-operative Oxford Knee Scores (OKS) was available for 5,268 patients between 1st April 2009 to 31st December 2016. See Fig 1 for the flow diagram of our full inclusion criteria.

### Main exposure variables

Our pre-operative exposures of interest were: 1) prior IASI and 2) previous minor knee arthroscopic surgery (diagnostic arthroscopy, joint lavage, meniscal repair, partial meniscectomy) with the same laterality as the index knee replacement procedure (identified using OPCS-4 laterality codes). All previous IASI were identified using READ codes and KA were identified using the classification of surgical operations and procedures version 4 (OPCS-4) codes in the procedure fields of the data set. Patients who had not received an IASI or KA prior to their index primary knee replacement were used as comparators. IASI and KA were analysed separately.

### Outcomes

Our primary outcomes of interest were subsequent revision and reoperation surgery following primary knee replacement for patients with prior IASI and KA. Secondary outcomes analysed included: timing to primary knee replacement (total or unicompartmental), defined as the time interval from diagnosis of osteoarthritis until the date of index primary knee replacement (total or unicompartmental); all-cause re-operation on the same joint; change in pre and 6-month post-operative Oxford Knee Score (OKS); 90 day and 1 year mortality; and post-

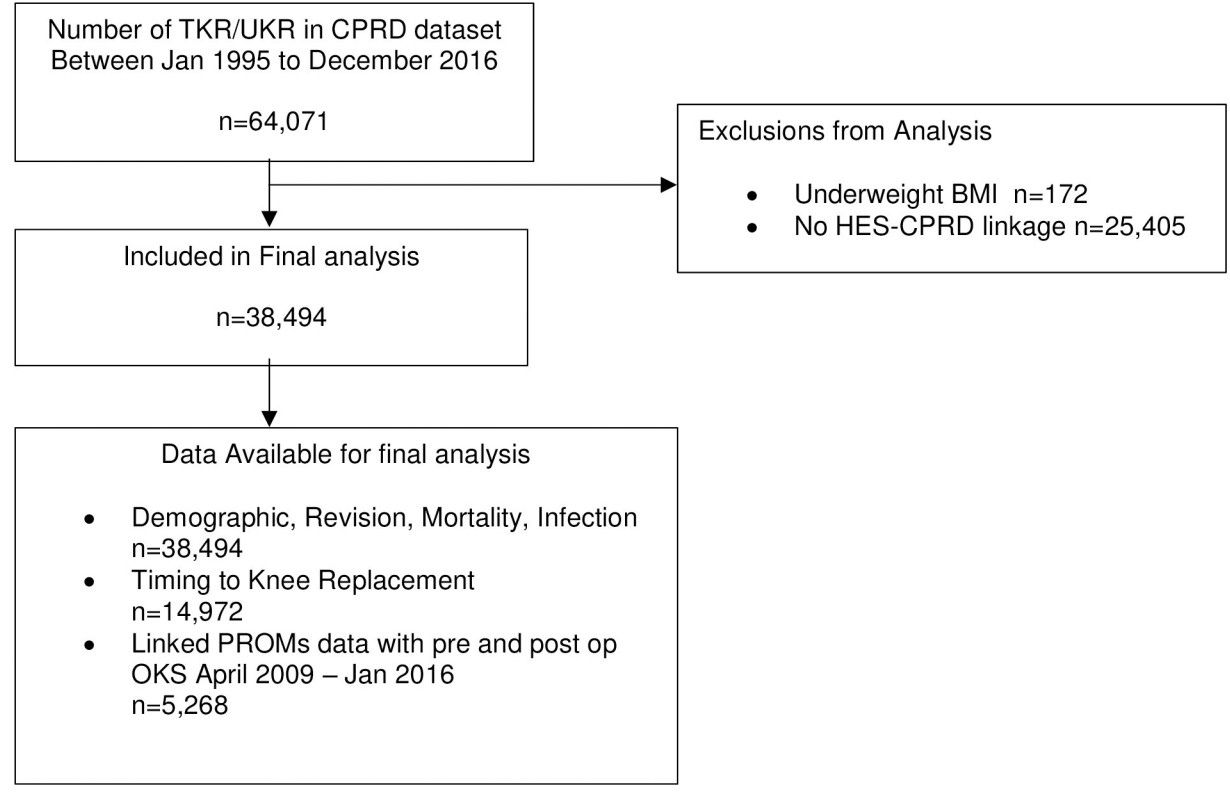

**Fig 1. Flow diagram outlining selection criteria for patients in the study.**

operative surgical site infection (SSI) at 3 months. We used 3 months as a cut off time point for SSI given that outcomes within this period are more likely to be directly related to knee replacement surgery.

## Confounders

Confounding factors to adjust for in regression models included: age (continuous), sex (Male, Female), smoking status (Ex, No, Yes), alcohol consumption (Ex, No, Yes), IMD score (quintiles of Least deprived, 2, 3, 4, Most deprived), Charlson Comorbidity Index (None, 1, 2, 3, 4+) and BMI (Normal, Overweight, Obese Class I, Obese Class II, Obese Class III). These confounding factors were selected based on their previously established roles as risk factors for the outcomes evaluated based on evidence from previous research [28].

## Ethical approval and informed consent

No information that can identify a patient is ever sent to CPRD from the contributing GP practices. Because a patient can't be identified from data a GP practice sends to CPRD, the GP practice doesn't need to seek a patient's consent to share data with CPRD (https://www.cprd. com/safeguarding-patient-data). CPRD has obtained ethical approval from a National Research Ethics Service Committee (NRES) for all purely observational research using anonymised CPRD data; namely, studies which do not include patient involvement. The study has been approved by ISAC (Independent Scientific Advisory Committee) for MHRA Database Research) (protocol number 17_127R). CPRD data linked to inpatient HES received on 9th May 2017; linked PROMs data on 25th July 2017. All data were fully anonymized before being

accessed for this study by the authors. The authors had no access to information that could identify individual participants.

## Statistical analysis

Stata v16.1 (StataCorp, Texas, USA) was used to perform statistical analysis. Descriptive statistics were used to summarise demographic characteristics: mean (SD) or median (IQR) for continuous variables and counts (percentages) for categorical variables. We analysed IASI and KA as a binary exposure prior to primary knee replacement using the date of most recent injection and arthroscopy prior to the surgery. We did not adjust our analysis to account for patients undergoing repeated exposures, or those exposed to both IASI and KA.

Linear regression was used to estimate beta coefficients (95% confidence intervals, CIs) for the relationships of previous IASI and KA with continuous outcomes of timing to index primary knee replacement (total and unicompartmental) and change in OKS. Cox proportional hazards regression analysis was used to calculate crude and multivariable-adjusted hazard ratios (HRs) with 95% CIs of undergoing revision surgery and re-operation following primary knee replacement with prior IASI or KA, after confirming no major departure from the assumptions of proportionality of hazards using Schoenfeld residuals. Cumulative 15-year survival probability for revision surgery was calculated using Kaplan-Meier estimates for patients receiving prior IASI and KA.

Logistic regression was used to estimate odds ratios (ORs) with 95% CIs of prior IASI and KA for post-operative OKS as a binary variable (improved and not improved), 90 day and 1 year mortality, and post-operative SSI at 3 months. Patients with a post-op OKS >4 were categorised as improved and patients with worse post-op scores or a score ≤4, were categorised as not improved. This reflects the minimal detectable change (MDC) of 4 points for OKS as per Beard et al. [29]. Due to the increased revision rates seen in primary unicompartmental knee replacements (UKRs), we conducted a sensitivity analysis which involved excluding all UKRs and patients with missing data for type of primary procedure and rerunning our regression models to assess the robustness of our results.

For all regression models we fitted: (model 1) unadjusted and (model 2) adjusted for confounding factors of: age, sex, smoking status, alcohol consumption, IMD score, Charlson Comorbidity Index and BMI. Complete case analyses were undertaken in adjusted regression models excluding patients with missing data on confounders.

## Results

### Prior intra-articular steroid injections and outcomes

A total of 12,703 (33%) patients received an IASI prior to undergoing their primary knee replacement vs. 25,791 (67%) that did not. The mean (SD) age of patients receiving an IASI was 70.1 (9.3) vs. 69.0 (9.8) for those that did not. Of the patients that received prior IASI, 59.5% were female and 40.5% were male. The demographic characteristics of patients receiving IASI prior to knee replacement are presented in Table 1.

The median timing from first diagnosis of OA in the index knee to primary knee replacement for patients receiving prior IASI was 2.5y (IQR 4.8) vs. 2.0y (IQR 4.1) for those that did not (Table 2).

The 15-year implant survival was lower in patients receiving prior IASI vs. those that did not (88.9% [95% CI 86.6–90.9] vs. 90.6% [95%CI 89.3–91.7]) (Table 3). The Kaplan Meier curves displayed in Fig 2 demonstrate a small, but consistent divergence over time that indicates a lower 15-year survival probability for revision for patients receiving prior IASI compared to those that did not (Fig 2). In unadjusted analysis, patients receiving prior IASI had a

**Table 1. Table of descriptive statistics for patients receiving intra-articular steroid injection and knee arthroscopy prior to primary knee replacement (TKR/UKA).**

| | Previous Intra-articular Steroid injection | | Previous Knee Arthroscopy | |
|---|---|---|---|---|
| | **Yes** | **No** | **Yes** | **No** |
| | **n (%)** | **n (%)** | **n (%)** | **n (%)** |
| | 12,703 (33) | 25,791 (67) | 6,347 (16.7) | 31,519 (83.2) |
| *total* | | 38,494 | | 37,866 |
| *missing* | | 0 | | 628 |
| **Confounder** | | | | |
| Male | 5,151 (40.5) | 11,499 (44.6) | 2,830 (44.6) | 13,520 (42.9) |
| Female | 7,552 (59.5) | 14,292 (55.4) | 3,517 (55.4) | 17,999 (57.1) |
| Age | | | | |
| mean (sd) | 70.1 (9.3) | 69.0 (9.8) | 64.6 (9.1) | 70.3 (9.4) |
| <60 | 2,559 (13.66) | 7,592 (16.81) | 1,835 (28.9) | 4,046 (12.8) |
| 60 to 70 | 6,044 (32.26) | 15,055 (33.34) | 2,575 (40.6) | 9,806 (31.1) |
| 70 to 79 | 7,055 (37.65) | 16,389 (36.29) | 1,594 (25.1) | 12,441 (39.5) |
| 80+ | 3,079 (16.43) | 6,126 (13.56) | 343 (5.4) | 5,226 (16.9) |
| BMI | | | | |
| Normal | 1,738 (15.9) | 3,486 (17.2) | 700 (13.1) | 4,432 (17.5) |
| Overweight | 4,117 (37.7) | 7,932 (39.2) | 1,981 (37.1) | 9,888 (39.1) |
| Obese Class I | 3,094 (28.3) | 5,522 (27.3) | 720 (13.5) | 6,838 (27.0) |
| Obese Class II | 1,421 (13.0) | 2,373 (11.7) | 720 (13.5) | 3,029 (12.0) |
| Obese Class III | 528 (4.8) | 909 (4.5) | 279 (5.2) | 1,135 (4.5) |
| Missing | 1,805 | 5,569 | 1,009 | 6,197 |
| Smoker | | | | |
| Ex | 4,173 (36.2) | 7,489 (33.6) | 2,070 (36.0) | 9,410 (43.2) |
| No | 6,342 (52.1) | 12,722 (57.1) | 3,037 (52.8) | 15,740 (57.3) |
| Yes | 991 (8.6) | 2,033 (9.1) | 643 (11.2) | 2,338 (8.5) |
| Missing | 1,197 | 3,547 | 597 | 4,031 |
| Alcohol Consumption | | | | |
| Ex | 299 (3.0) | 477 (2.6) | 128 (2.7) | 630 (2.8) |
| No | 1,820 (18.7) | 3,153 (17.5) | 778 (16.6) | 4,100 (18.1) |
| Yes | 7,599 (78.2) | 14,377 (79.8) | 3,770 (80.6) | 17,896 (79.1) |
| Missing | 2,985 | 7,784 | 1,671 | 8,893 |
| Deprivation Index Rank | | | | |
| Least | 2,913 (22.9) | 6,560 (25.4) | 1,487 (23.4) | 7,836 (24.9) |
| 2 | 2,886 (22.7) | 6,336 (24.5) | 1,516 (23.9) | 7,553 (24.0) |
| 3 | 2,995 (23.2) | 5,525 (21.4) | 1,436 (22.6) | 6,919 (22.0) |
| 4 | 2,254 (17.7) | 4,268 (16.5) | 1,120 (17.7) | 5,298 (16.8) |
| Most | 1,686 (13.2) | 3,073 (11.9) | 787 (12.4) | 3,876 (12.3) |
| Missing | 9 | 29 | 1 | 37 |
| Charlson Score | | | | |
| None | 8,530 (67.1) | 18,638 (72.2) | 4,714 (74.3) | 22,063 (70.0) |
| 1 | 1,451 (11.4) | 2,511 (9.7) | 572 (9.0) | 3,303 (10.5) |
| 2 | 1,502 (11.8) | 2,594 (10.0) | 558 (8.8) | 3,461 (11.0) |
| 3 | 506 (3.9) | 902 (3.50) | 216 (3.4) | 1,161 (3.7) |
| 4+ | 714 (5.6) | 1,146 (4.4) | 287 (4.5) | 1,531 (4.7) |

**Table 2. Summary table for outcomes for patient receiving intra-articular steroid injection and knee arthroscopy prior to knee replacement (TKR/UKR).**

| Outcome | Previous IASI | | Previous KA | |
|---|---|---|---|---|
| | **Yes** | **No** | **Yes** | **No** |
| | **n (%)** | **n (%)** | **n (%)** | **n (%)** |
| | 12,703 (33) | 25,791 (67) | 6,347 (16.7) | 31,519 (82.3) |
| Timing from first Diagnosis of OA to Primary Replacement (yr) | | | | |
| <1 | 1,490 (25.4) | 2,876 (31.6) | 584 (24.2) | 3,711 (30.1) |
| 1 to 2 | 1,021 (17.4) | 1,667 (18.3) | 463 (19.2) | 2,183 (17.7) |
| 2 to 3 | 701 (11.9) | 1,039 (11.4) | 314 (13.1) | 1,394 (11.3) |
| 3 to 4 | 506 (8.6) | 750 (8.2) | 235 (9.7) | 1,002 (8.1) |
| 4 to 5 | 432 (7.4) | 542 (6) | 162 (6.7) | 789 (6.4) |
| 5 to 10 | 1,148 (19.5) | 1,597 (17.6) | 462 (19.2) | 2,241 (18.1) |
| 10+ | 576 (9.8) | 627 (6.9) | 184 (7.6) | 1,001 (8.1) |
| missing | 6,829 | 16,693 | 3,943 | 19,189 |
| Median (IQR) | 2.5 (4.8) | 2.0 (4.1) | 2.4 (4.3) | 2.1 (4.4) |
| 90 Day Mortality | 27 (0.2) | 63 (0.2) | 3 (0.05) | 86 (0.3) |
| 1y Mortality | 126 (1.0) | 276 (1.1) | 21 (0.3) | 370 (1.2) |
| Revision following Primary Replacement | 321 (2.5) | 642 (2.5) | 403 (6.3) | 560 (1.7) |
| Reoperation following Primary Replacement | 706 (5.5) | 1,339 (5.2) | 898 (14.1) | 1,147 (3.6) |
| Post-operative Infection within 3m | 13 (0.10) | 9 (0.03) | 3 (0.05) | 18 (0.06) |
| Change in OKS following Primary Procedure | | | | |
| No. Observations | 1,972 | 3,296 | 1,585 | 3,683 |
| Mean (SD) | 15.7 (9.9) | 16 (9.7) | 14.7 (10.1) | 16.4 (9.5) |

**Table 3. Survival probability for revision at 1,5,10,15 years for patients receiving prior intra articular steroid injection (IASI) and knee arthroscopy (KA).**

| Time (years) | Survival probability (%) | standard error | 95% CI |
|---|---|---|---|
| Revision Risk Without Prior IASI | | | |
| 1 | 99.6 | 0.04 | 99.5 to 99.7 |
| 5 | 98.4 | 0.09 | 98.2 to 98.6 |
| 10 | 95.9 | 0.20 | 95.5 to 96.3 |
| 15 | 90.6 | 0.61 | 89.3 to 91.7 |
| Revision Risk With Prior IASI | | | |
| 1 | 99.6 | 0.06 | 99.4 to 99.7 |
| 5 | 98.0 | 0.15 | 97.7 to 98.3 |
| 10 | 94.8 | 0.37 | 94.0 to 95.5 |
| 15 | 88.9 | 1.07 | 86.6 to 90.9 |
| Revision Risk Without Prior KA | | | |
| 1 | 99.7 | 0.03 | 99.7 to 99.8 |
| 5 | 98.8 | 0.07 | 98.7 to 99.0 |
| 10 | 97.1 | 0.16 | 96.7 to 97.4 |
| 15 | 92.4 | 0.53 | 91.3 to 93.4 |
| Revision Risk With Prior KA | | | |
| 1 | 98.7 | 0.14 | 98.4 to 99.0 |
| 5 | 95.1 | 0.33 | 94.4 to 95.7 |
| 10 | 87.2 | 0.76 | 85.6 to 88.6 |
| 15 | 75.6 | 2.21 | 70.9 to 79.6 |

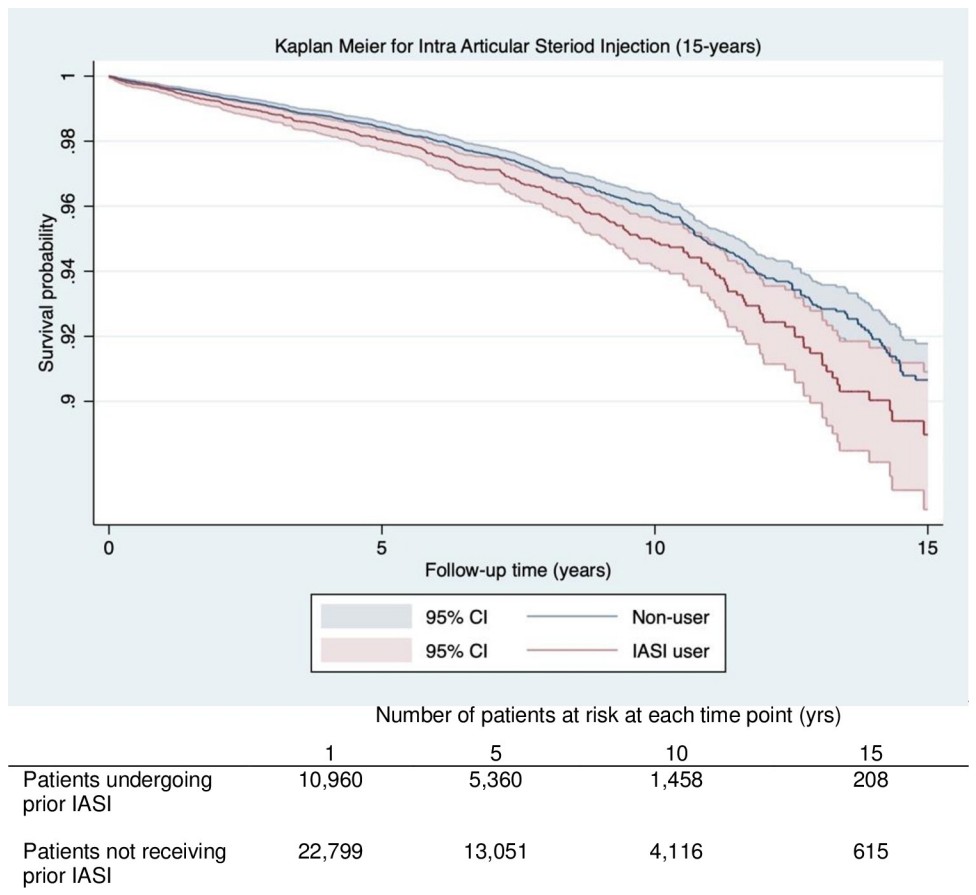

**Fig 2. Kaplan Meier graph for survival probability (Y axis) of revision surgery over time (X axis) for patients receiving prior intra-articular steroid injections (IASI) (Red) vs non-users (Blue) following knee replacement.** The number of patients at risk for at each time point (1,5,10,15) are displayed in the table below the graph.

22% increased risk of revision (HR = 1.22 95%CI (1.07 to 1.40), which increased to 25% on adjusting for potential confounders (HR = 1.25 95%CI (1.06 to 1.49)) (Table 4).

   Survival probability for re-operation on the same joint and the same limb following the index knee replacement procedure at 15 years was lower in patients receiving IASI prior to primary KA vs. those without (90.7% [95%CI 89.4–91.9] vs. 92.2% [95%CI 91.5–92.9]) (Table 5). Unadjusted analysis showed a 16% increased risk of re-operation with prior IASI (HR = 1.16 95%CI (1.06 to 1.27)), which was 18% following adjustment for potential confounders (HR = 1.18 95%CI (1.05 to 1.32)) (Table 4). The Kaplan Meier curves displayed in Fig 3 demonstrate a small divergence that remains consistent with time and indicates a lower 15-year survival probability for re-operation in patients receiving prior IASI compared to those that did not (Fig 3).

   For patients receiving IASI prior to primary knee replacement, the mean (SD) change in OKS (i.e. post-operative OKS–preoperative OKS) was 15.6 (9.9) and in patients without prior IASI the mean (SD) change in OKS was 16.0 (9.6) (Table 2). Unadjusted and multivariable adjusted linear regression analyses showed no associations with change in OKS for patients receiving prior IASI, adjusted mean difference -0.02 95%CI (-0.88 to 0.34) (Table 6). Furthermore, prior use of IASI was not associated with clinically meaningful change in OKS as a binary variable, odds ratio 1.02 95%CI (0.84 to 1.23) (Table 7).

**Table 4. Hazard ratios from the Cox proportional regression models for revision risk and re-operation risk with prior intra-articular steroid injection (IASI) and knee arthroscopy (KA) following knee replacement.**

| Outcome | Haz. Ratio (95%CI) | P Value |
|---|---|---|
| Revision Risk with prior IASI | | |
| Crude | 1.22 (1.07 to 1.40) | 0.003 |
| Adjusted (without BMI) | 1.29 (1.09 to 1.53) | 0.002 |
| Adjusted with BMI | 1.25 (1.06 to 1.49) | 0.009 |
| Revision Risk with prior KA | | |
| Crude | 4.22 (3.77 to 4.89) | <0.001 |
| Adjusted (without BMI) | 3.22 (2.73 to 3.18) | <0.001 |
| Adjusted with BMI | 3.14 (2.64 to 3.73) | <0.001 |
| Re-operation Risk with prior IASI | | |
| Crude | 1.16 (1.06 to 1.27) | 0.002 |
| Adjusted (without BMI) | 1.20 (1.07 to 1.34) | 0.001 |
| Adjusted with BMI | 1.18 (1.05 to 1.32) | 0.005 |
| Re-operation Risk with prior KA | | |
| Crude | 4.39 (4.00 to 4.82) | <0.001 |
| Adjusted (without BMI) | 3.39 (3.03 to 3.80) | <0.001 |
| Adjusted with BMI | 3.25 (2.89 to 3.66) | <0.001 |

A total of 13 patients (0.1%) who received prior IASI developed post-operative SSI (Table 2). Unadjusted and adjusted logistic regression models showed that prior use of IASI increased the odds of infection by about 3-fold ([unadjusted OR 2.93 95%CI (1.25 to 6.87)] & [adjusted OR 3.10 95%CI (1.14–8.46)]) (Table 7).

**Table 5. Survival probability for reoperation same limb, same site at 1,5,10,15 years for patients receiving prior intra articular steroid injection (IASI) and knee arthroscopy (KA).**

| Time (years) | Survival probability (%) | standard error | 95% CI |
|---|---|---|---|
| Re-operation Risk Without Prior IASI | | | |
| 1 | 98.0 | 0.09 | 97.8 to 98.2 |
| 5 | 95.3 | 0.15 | 95.0 to 95.5 |
| 10 | 93.6 | 0.20 | 93.2 to 94.0 |
| 15 | 92.2 | 0.35 | 91.5 to 92.9 |
| Re-operation Risk With Prior IASI | | | |
| 1 | 97.8 | 0.13 | 97.5 to 98.0 |
| 5 | 94.5 | 0.23 | 94.0 to 94.9 |
| 10 | 92.3 | 0.35 | 91.6 to 93.0 |
| 15 | 90.7 | 0.64 | 89.4 to 91.9 |
| Re-operation Risk Without Prior KA | | | |
| 1 | 98.6 | 0.07 | 98.4 to 98.7 |
| 5 | 96.7 | 0.11 | 96.4 to 96.9 |
| 10 | 95.6 | 0.15 | 95.3 to 95.9 |
| 15 | 94.6 | 0.27 | 94.1 to 95.1 |
| Re-operation Risk With Prior KA | | | |
| 1 | 94.7 | 0.29 | 94.1 to 95.2 |
| 5 | 86.4 | 0.49 | 85.5 to 87.4 |
| 10 | 81.0 | 0.70 | 79.6 to 82.4 |
| 15 | 75.9 | 1.58 | 72.6 to 78.8 |

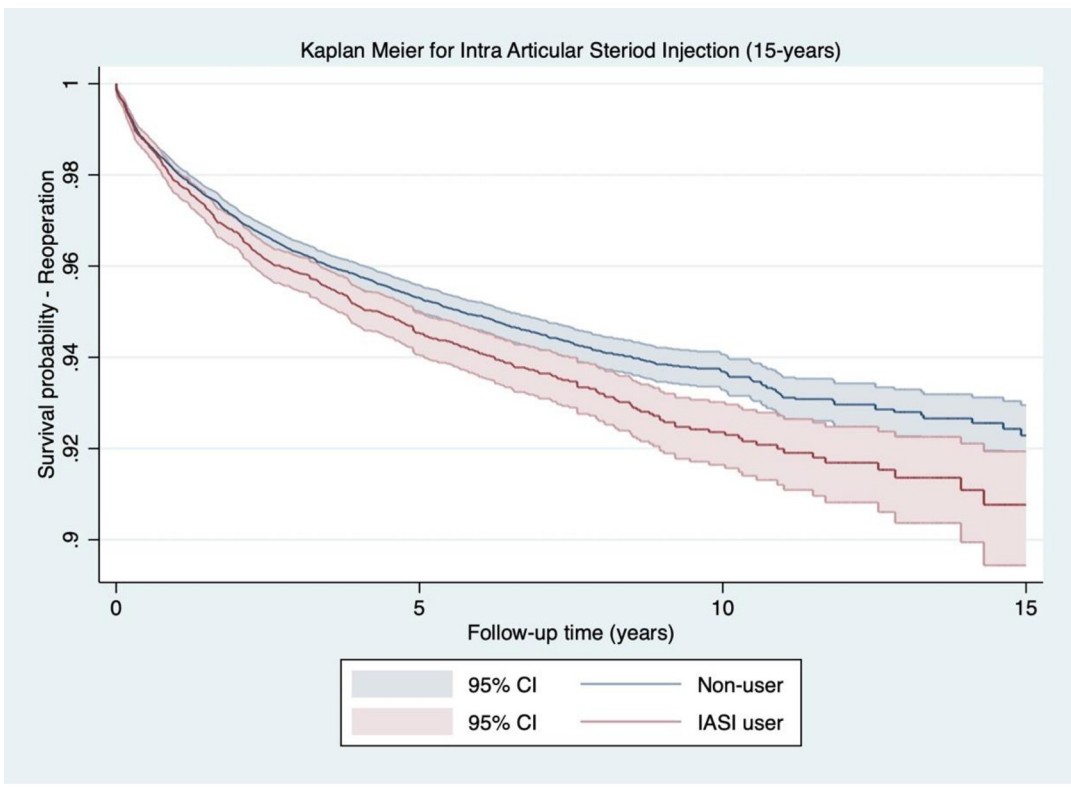

Kaplan Meier for Intra Articular Steriod Injection (15-years)

| | Number of patients at risk at each time point (yrs) | | | |
|---|---|---|---|---|
| | 1 | 5 | 10 | 15 |
| Patients undergoing prior IASI | 10,960 | 5,360 | 1,458 | 208 |
| Patients not receiving prior IASI | 22,799 | 13,051 | 4,116 | 615 |

**Fig 3. Kaplan Meier graph for survival probability (Y axis) of all cause re-operation over time (X axis) for patients receiving prior steroid injection (Red) vs patients with no prior steroid injection (Blue) following knee replacement.** The number of patients at risk for at each time point (1,5,10,15) are displayed in the table below the graph.

There was no evidence of an association with 90-day mortality with prior use of IASI (Table 7). However, prior IASI was associated with a reduction in the odds of 1-year mortality (OR = 0.76 95%CI (0.58 to 1.00)) (Table 7).

In sensitivity analyses which involved excluding UKRs, similar associations were observed, with an adjusted revision risk HR 1.17 95%CI (0.97 to 1.41). For re-operation, the adjusted risk HR 1.14 95%CI (1.00 to 1.29) (Table 8).

## Knee arthroscopy and outcomes

A total of 6,347 (16.5%) patients underwent KA prior to primary knee replacement vs. 31,519 (83.2%) that did not. The mean (SD) age of patients who received KA prior to primary knee replacement was 64.9 (9.1). Of the patients that received prior KA, 55.4% were female and 44.6% were male. The demographic characteristics of patients receiving KA prior to primary knee replacement are displayed in Table 1.

**Table 6. Linear regression models (crude, adjusted without BMI and fully adjusted) for timing to total knee replacement, change in OKS for patient receiving prior intra-articular steroid injection (IASI) and knee arthroscopy (KA).**

| Linear Regression Models | Coefficient (95% CI) | P-value |
|---|---|---|
| Timing to total knee replacement with prior IASI | | |
| Crude | 0.59 (0.47 to 0.72) | <0.001 |
| Adjusted (without BMI) | 0.61 (0.46 to 0.76) | <0.001 |
| Adjusted with BMI | 0.60 (0.45 to 0.75) | <0.001 |
| Timing to total knee replacement with prior KA | | |
| Crude | 0.12 (-0.04 to 0.28) | 0.146 |
| Adjusted (without BMI) | 0.31 (0.11 to 0.51) | 0.002 |
| Adjusted with BMI | 0.28 (0.08 to 0.49) | 0.006 |
| Change in OKS with prior IASI | | |
| Crude | -0.36 (-0.90 to 0.18) | 0.195 |
| Adjusted (without BMI) | -0.31 (-0.91 to 0.28) | 0.304 |
| Adjusted with BMI | -0.02 (-0.88 to 0.34) | 0.394 |
| Change in OKS with prior KA | | |
| Crude | -1.67 (-2.24 to -1.09) | <0.001 |
| Adjusted (without BMI) | -1.69 (-2.36 to -1.02) | <0.001 |
| Adjusted with BMI | -1.63 (-2.31 to -0.95) | <0.001 |

The median timing from first diagnosis of OA in the index knee to primary knee replacement was similar for patients receiving prior KA and those that did not: 2.4 (IQR 4.3) vs. 2.1 (IQR 4.4)) years (Table 2).

Patients undergoing KA prior to primary knee replacement had lower implant survival rates at 15 years vs. patients that did not undergo prior KA (75.6% [95%CI 72.6–78.8] vs 92.4% [95%CI 91.3–93.4] (Table 3). In unadjusted analysis, prior KA was associated with an increased risk of revision (HR = 4.22 95%CI (3.77 to 4.89)), which was slightly attenuated on multivariable adjustment (HR = 3.14 95%CI (2.64 to 3.73)) (Table 4). Kaplan Meier curves shown in Fig 4 diverge substantially over time, demonstrating a lower 15-year implant survival probability between patients receiving prior KA and those that did not (Fig 4).

Prior KA was associated with a 4-fold increase in the unadjusted risk of re-operation compared to patients that did not undergo prior KA (HR = 4.39 95%CI (4.00 to 4.82)), which was attenuated to 3-fold following adjustment for potential confounders (HR = 3.25 95%CI (2.89 to 3.66)) (Table 4). Kaplan Meier curves displayed in Fig 5 diverge substantially over time and indicating a lower 15-year survival probability for re-operation for patients receiving prior KA compared with those that did not (Fig 5).

For patients receiving prior KA, the mean (SD) change in OKS was lower than those that did not undergo KA: 14.7 (10.1) vs. 16.4 (9.5) (Table 2). Receiving prior KA was associated with a reduced change in OKS following knee replacement (unadjusted estimate: -1.67 95%CI (-2.24 to -1.09) and adjusted estimate: -1.63 95%CI (-2.31 to -0.95)) (Table 6). Patients receiving prior KA had a 65% increased unadjusted odds of not receiving clinically meaningful improvement following knee replacement (OR = 1.65 95%CI (1.40 to 1.96)), which was minimally attenuated on multivariable adjustment (OR = 1.56 95%CI (1.27 to 1.91)) (Table 7).

SSI rates in the immediate 3-month postoperative period were similar in patients who received prior KA and those that did not. There was no evidence of an association between prior KA and post-operative SSI in both unadjusted and multivariable adjusted analyses (Table 7).

**Table 7. Logistic regression models (crude, adjusted without BMI and fully adjusted) for 90 day & 1 year mortality, surgical site injection at 3 months, poor OKS outcome for patients receiving prior intra-articular steroid injection (IASI) and knee arthroscopy (KA) following knee replacement.**

| Logistic Regression Model | odds ratio (95% CI) | P-value |
|---|---|---|
| 90 Day Mortality with prior IASI | | |
| Crude | 0.87 (0.55 to 1.36) | 0.545 |
| Adjusted (without BMI) | 0.70 (0.39 to 1.25) | 0.234 |
| Adjusted with BMI | 0.67 (0.36 to 1.24) | 0.205 |
| 1 year Mortality with prior IASA | | |
| Crude | 0.92 (0.74 to 1.14) | 0.478 |
| Adjusted (without BMI) | 0.74 (0.57 to 0.96) | 0.028 |
| Adjusted with BMI | 0.76 (0.58 to 1.00) | 0.057 |
| 90 Day Mortality with prior KA | | |
| Crude | 0.17 (0.05 to 0.54) | 0.003 |
| Adjusted (without BMI) | 0.15 (0.02 to 1.11) | 0.064 |
| Adjusted with BMI | 0.16 (0.02 to 1.22) | 0.079 |
| 1 year Mortality with prior KA | | |
| Crude | 0.27 (0.17 to 0.43) | <0.001 |
| Adjusted (without BMI) | 0.43 (0.25 to 0.737) | 0.002 |
| Adjusted with BMI | 0.44 (0.25 to 0.76) | 0.003 |
| Surgical Site Infection at 3 months with prior IASI | | |
| Crude | 2.93 (1.25 to 6.87) | 0.013 |
| Adjusted (without BMI) | 3.26 (1.20 to 8.90) | 0.020 |
| Adjusted with BMI | 3.10 (1.14 to 8.46) | 0.027 |
| Surgical Site Infection at 3 months with prior KA | | |
| Crude | 0.82 (0.24 to 2.81) | 0.762 |
| Adjusted (without BMI) | 0.59 (0.13 to 2.69) | 0.502 |
| Adjusted with BMI | 0.61 (0.13 to 2.76) | 0.552 |
| Poor Oxford Knee Score Outcome with Prior IASI | | |
| Crude | 1.06 (0.89 to 1.25) | 0.490 |
| Adjusted (without BMI) | 1.05 (0.87 to 1.28) | 0.551 |
| Adjusted with BMI | 1.02 (0.84 to 1.23) | 0.830 |
| Poor Oxford Knee Score Outcome with Prior KA | | |
| Crude | 1.65 (1.40 to 1.96) | <0.001 |
| Adjusted (without BMI) | 1.54 (1.27 to 1.91) | <0.001 |
| Adjusted with BMI | 1.56 (1.27 to 1.91) | <0.001 |

Receiving prior KA was associated with a reduced odds of 90-day mortality (unadjusted OR = 0.17 95%CI (0.05 to 0.54) and adjusted OR = 0.16 95%CI 0.16 (0.02 to 1.22)) and 1-year mortality (unadjusted OR = 0.27 95%CI (0.17 to 0.43) and adjusted OR = 0.44 95%CI (0.25 to 0.76)) (Table 7).

In sensitivity analysis which involved excluding UKRs, there was attenuation in the adjusted risks for revision and re-operation, HR 1.70 95%CI (1.40 to 2.05) and HR 1.64 95%CI (1.44 to 1.87) respectively (Table 8).

## Discussion

Our results indicate that receiving IASI or KA prior to primary knee replacement was associated with higher rates of revision and re-operation. Receiving IASI was associated with a lower 15-year implant survival and a 3-fold increase in risk of SSI in the immediate 3-month post-

**Table 8. Sensitivity analysis removing uni-compartmental knee replacement (UKR) and patients with missing data for primary procedure type.**

| Outcome | Haz. Ratio (95%CI) | P Value |
|---|---|---|
| Revision Risk with prior IASI | | |
| Crude | 1.16 (1.00 to 1.34) | 0.046 |
| Adjusted (without BMI) | 1.19 (0.99 to 1.43) | 0.054 |
| Adjusted with BMI | 1.17 (0.97 to 1.41) | 0.093 |
| Revision Risk with prior KA | | |
| Crude | 2.21 (1.91 to 2.54) | <0.001 |
| Adjusted (without BMI) | 1.75 (1.45 to 2.10) | <0.001 |
| Adjusted with BMI | 1.70 (1.40 to 2.05) | <0.001 |
| Re-operation Risk with prior IASI | | |
| Crude | 1.12 (1.01 to 1.24) | 0.028 |
| Adjusted (without BMI) | 1.16 (1.03 to 1.31) | 0.013 |
| Adjusted with BMI | 1.14 (1.00 to 1.29) | 0.039 |
| Re-operation Risk with prior KA | | |
| Crude | 2.28 (2.06 to 2.52) | <0.001 |
| Adjusted (without BMI) | 1.71 (1.51 to 1.51) | <0.001 |
| Adjusted with BMI | 1.64 (1.44 to 1.87) | <0.001 |

operative period following knee replacement, although the event rate was low making interpretation difficult. Timing from diagnosis of OA to knee replacement was a median of 6 months longer in patients receiving IASI. Knee arthroscopy was associated with a lower clinically meaningful improvement in 6-month OKS and 15-year implant survival following primary knee replacement. Both interventions were associated with a lower risk of 1-year mortality following primary knee replacement, for which there is not a biologically plausible mechanism, suggesting potential for selection bias due to confounding by indication for patients receiving these interventions.

Based on robust evidence from a randomised controlled trial (RCT) and a systematic review, a recent clinical practice guideline published in the British Medical Journal recommended against the use of arthroscopy in treating degenerative knee disease [21]. The RCT found that KA was no more effective than exercise therapy in a middle-aged population with degenerative meniscal tears that had no radiographical evidence of OA [30]. In a meta-analysis of nine RCTs that compared KA with placebo and non-surgical treatment in patients with degenerative knees, only a small improvement in knee pain was observed in the surgically treated patients, and this did not persist beyond a year [31]. Some of our findings are consistent with those of other observational studies that have evaluated the impact of prior KA on outcomes following knee replacement [24, 25, 32]. In a large study of 138,019 patients, KA performed within 2 years prior to knee replacement was associated with an increased risk of revision [25]. Piedade et al., utilising a cohort of 1474 patients, reported that prior KA had a higher postoperative complication rate (reoperations and revision), higher failure and worse survival following knee replacement [24]. Other studies based on small sample sizes have not demonstrated significant differences in complication rates and this is likely because they are underpowered to detect any differences [26, 27]. We also observed worse clinical outcome in patients undergoing KA prior to knee replacement. Patients receiving prior KA had 1.6 points less change in OKS. Whilst this may have been statistically significant, it is less than the MCID of 4 that would indicate a true clinical difference between the groups [29]. When defining outcome as a binary variable of whether or not patients achieved a clinically meaningful

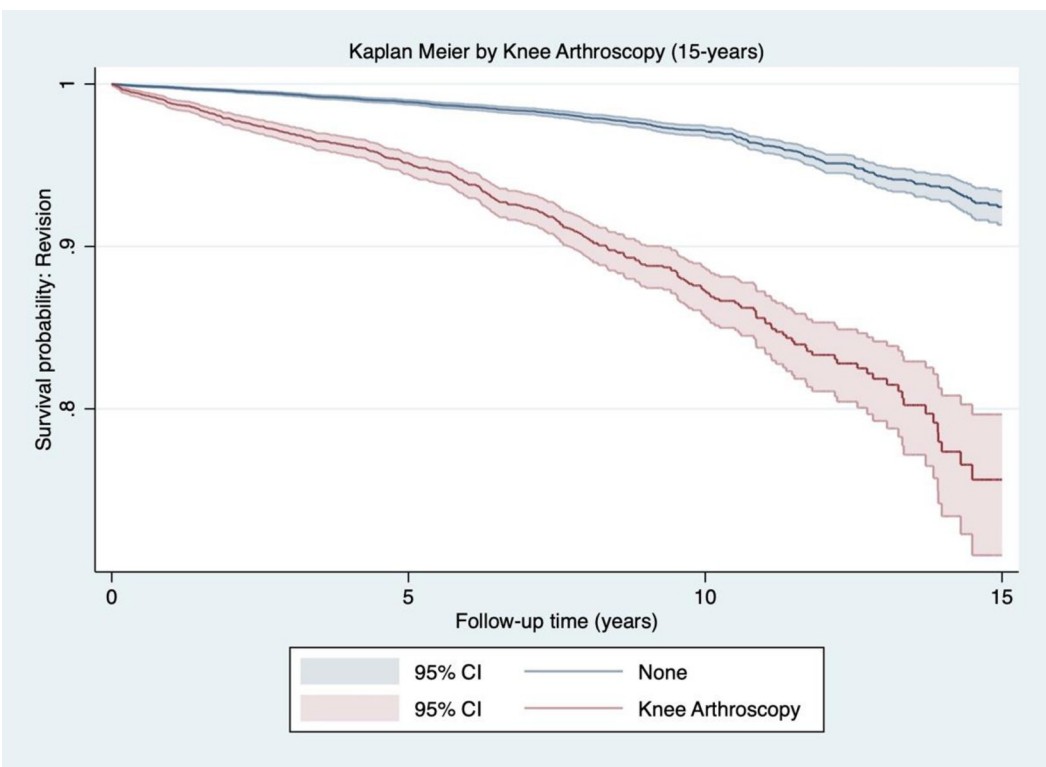

Kaplan Meier by Knee Arthroscopy (15-years)

| | Number of patients at risk at each time point (yrs) | | | |
|---|---|---|---|---|
| | 1 | 5 | 10 | 15 |
| Patients undergoing prior KA | 5,489 | 2,739 | 751 | 59 |
| Patients not receiving prior KA | 27,688 | 15,326 | 4,725 | 740 |

**Fig 4. Kaplan Meier graph for survival probability (Y axis) of revision surgery over time (X axis) for patients receiving prior knee arthroscopy (Red) vs patients with no prior knee arthroscopy (Blue) following knee replacement.** The number of patients at risk for at each time point (1,5,10,15) are displayed in the table below the graph.

improvement, KA was associated with a 65% increased risk of not achieving clinically meaningful improvement. Previous literature is inconclusive regarding prior KA and its effect on post-operative PROMS. Several small cohort studies have not shown a difference in post-operative PROMs in patients undergoing KA prior to knee replacement [24, 26, 27]. One study reported a reduction in post-operative OKS compared to their control group when KA was performed within 6 months of primary joint replacement (32.8 vs. 36.3) [32]. In addition to evaluating revision and re-operation outcomes, we reported on outcomes such as timing to knee replacement, implant survival, infection, patient reported outcome (OKS), and mortality. Furthermore, our study population comprised both unicompartmental and total knee replacements. Unicompartmental knee replacements have been shown to be associated with increased revision risk [33, 34]. Our sensitivity analysis demonstrated little effect on the HR for revision and reoperation risks with prior IASI, however, there was a large attenuation in the risks for revision and reoperation risks with prior KA.

Intra-articular steroid injection is recommended by NICE as an adjunct to core non-surgical treatment for OA prior to knee replacement. Recent evidence has observed high patient

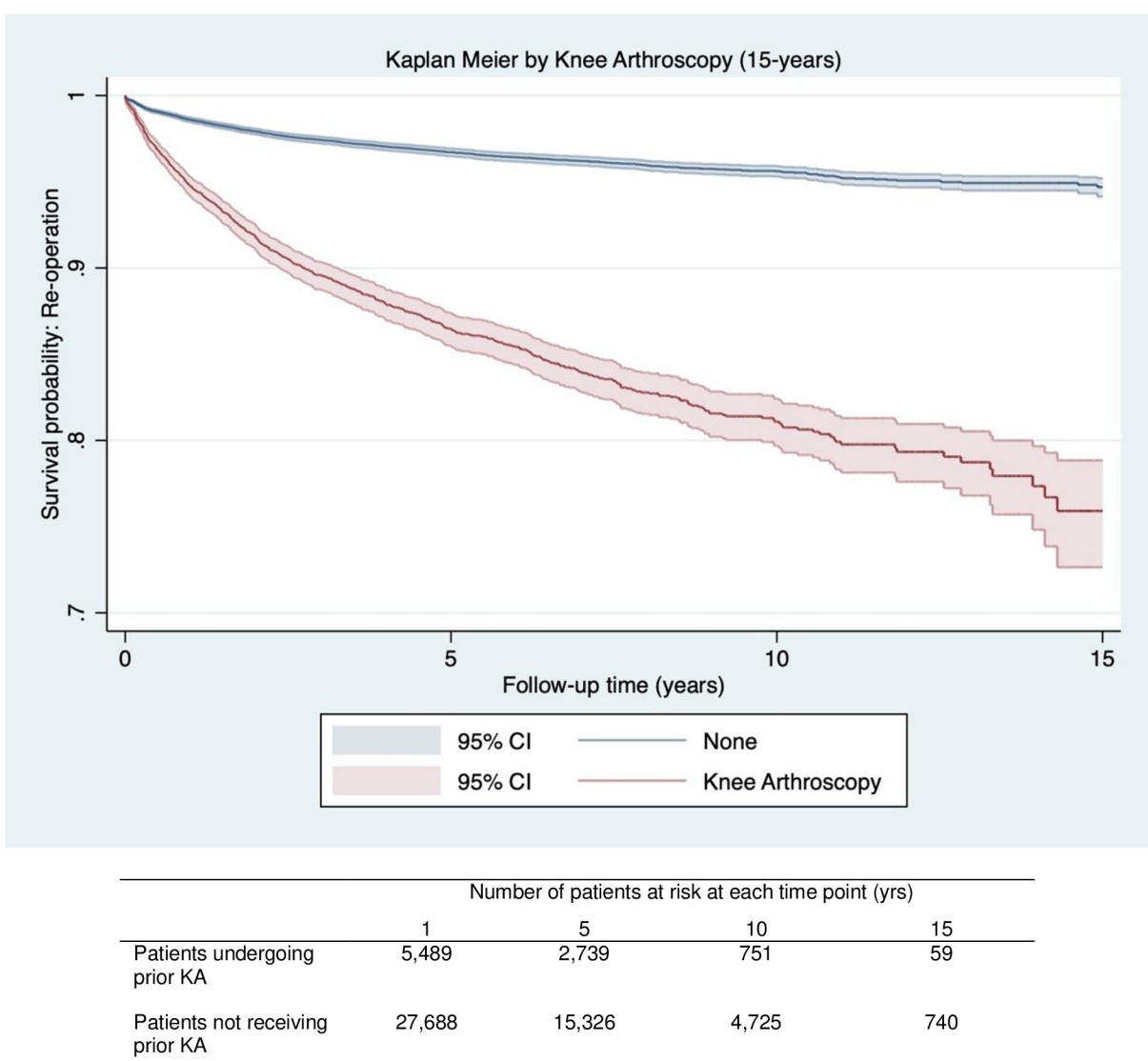

| | Number of patients at risk at each time point (yrs) | | | |
|---|---|---|---|---|
| | 1 | 5 | 10 | 15 |
| Patients undergoing prior KA | 5,489 | 2,739 | 751 | 59 |
| Patients not receiving prior KA | 27,688 | 15,326 | 4,725 | 740 |

**Fig 5. Kaplan Meier graph for survival probability (Y axis) of all cause re-operation over time (X axis) for patients receiving prior knee arthroscopy (Red) vs patients with no prior knee arthroscopy (Blue) following knee replacement.** The number of patients at risk for at each time point (1,5,10,15) are displayed in the table below the graph.

satisfaction rates with steroid injections when treating knee pain in OA, however, less than 50% of patients receive IASI prior to surgery [35]. Consistent with the literature [7], we observed that 33% of patients received IASI prior to knee replacement. There are a number of reports on the impact of prior IASI on outcomes following knee replacement, but the majority have reported infection outcomes. A meta-analysis of observational studies published in 2014 did not observe any statistically significant risk of deep or superficial infection with prior IASI, but cited small sample sizes as a limiting factor [36]. A recent large retrospective cohort of 35,890 patients reported an increased risk of deep and superficial infection with prior IASI when given within 3 to 6 months before total knee replacement; however, the risk for infection was not significant when injections were given more than 6 months before surgery [10]. Our findings demonstrated a 3-fold increased risk of SSI following knee replacement with prior IASI, although the event rate was low making interpretation difficult. The 6-month delay for

surgery seen in patients with prior IASI may reflect surgeons' response to previously published work, which reported increased risk of infections when steroid injections were administered less than 6 months before surgery. To our knowledge the current study is the first to report the relationships of prior IASI with revision and reoperation outcomes following primary knee replacement.

It is possible the observed reduction in 1-year mortality associated with prior KA is the result of confounding by indication. Patients who had undergone prior KA were on average 5.7 years younger at the time of primary knee replacement than those that did not undergo prior KA (64.6 vs 70.3) (Table 1). Younger age at primary knee replacement has been associated with lower risk of mortality [37]. It may also explain the higher revision rates seen in the KA group, younger patients at time of primary knee replacement are associated with increased life time risk of revision [38]. Prior IASI was associated with reduced odds of 1-year mortality (OR = 0.76 95%CI (0.58 to 1.00)), however much less so than prior KA, this may be due to more balanced demographics seen in prior IASI users and non-users especially age (Table 1). There is further potential for residual confounding that may be responsible for the reduction in OR for mortality in KA and IASI users. There may be important unknown and unmeasured confounders that where not accounted and controlled for in our analysis.

It should be acknowledged that patients undergoing prior IASI and KA are not two mutually exclusive groups. We calculated that within our study population 2,307 patients underwent both IASI and KA prior to primary knee replacement (S1 Table). This could introduce bias from treatment cross over, however the overall number of patients receiving both interventions is low. Our studies primary research question focused on whether prior IASI and KA is associated with timings and outcomes of surgery, and thus we have not reported on patients undergoing both interventions as a further exposure group. To our knowledge there does not appear to be any prior published literature that has investigated the impact of undergoing both KA and IASI prior to primary knee replacement and this could warrant further investigation in future studies.

A strength of our study was the use of a large-scale dataset comprising a study population that is representative of the general population of England [39]. The study population also closely reflects that of the National Joint Registry of England and Wales with comparable patient demographics on age, sex and numbers of total knee replacements and UKR. We compared the characteristics of patients in the full CPRD GOLD versus CPRD GOLD-HES linked datasets, and they were similar with respect to confounding factors with no evidence of responder bias (S2 Table). We employed a comprehensive list of relevant clinical outcomes that are important to both patients and clinicians. Finally, we conducted sensitivity analyses to confirm the robustness of the results. The limitations deserve mention, and these include (i) the use of an observational design which is limited by residual confounding, and inability to prove causation and (ii) inability to account for patients that may have received both prior IASI and KA, which presents a potential bias due to treatment crossover.

## Conclusion

Steroid injections and knee arthroscopy prior to primary knee replacement are each associated with worse outcomes. The observed association of lower mortality for patients receiving these interventions is suggestive of confounding by indication. Surgeons should be mindful of the associated risks when recommending IASI or KA to patients who are on the treatment pathway for knee osteoarthritis and counsel these patients appropriately as part of the shared decision-making process.

## Supporting information

**S1 Table. Table displaying the number of patients within our study population that underwent both intra-articular steroid injection (IASI) and knee arthroscopy (KA) prior to primary knee replacement.**
(DOCX)

**S2 Table. Characteristics of patients in the full CPRD versus CPRD-HES linked datasets.**
(DOCX)

## Acknowledgments

This study is based in part on data from the CPRD obtained under licence from the UK Medicines and Healthcare products Regulatory Agency. However, the interpretation and conclusions contained in this study are those of the author/s alone.

## Author Contributions

**Conceptualization:** Matthew Strang, John Broomfield, Michael Whitehouse, Antonella Delmestri, Andrew Judge.

**Data curation:** Antonella Delmestri.

**Formal analysis:** Matthew Strang, Andrew Judge.

**Methodology:** Setor Kunutsor, Antonella Delmestri, Ashley Blom, Andrew Judge.

**Project administration:** Andrew Judge.

**Supervision:** Michael Whitehouse, Ashley Blom, Andrew Judge.

**Writing – original draft:** Matthew Strang, John Broomfield, Michael Whitehouse, Setor Kunutsor, Sion Glyn-Jones, Antonella Delmestri, Ashley Blom, Andrew Judge.

**Writing – review & editing:** Matthew Strang, John Broomfield, Michael Whitehouse, Setor Kunutsor, Sion Glyn-Jones, Antonella Delmestri, Ashley Blom, Andrew Judge.

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
