## [Decision Letter · Decision Letter 0]

2 Aug 2024

PONE-D-24-25859Association of intra-articular injection and knee arthroscopy prior to primary knee replacement with the timing and outcomes of surgery: Retrospective cohort study using data from the Clinical Practice Research Datalink GOLD databasePLOS ONE

Dear Dr. Judge,

Thank you for submitting your manuscript to PLOS ONE. After careful consideration, we feel that it has merit but does not fully meet PLOS ONE’s publication criteria as it currently stands. Therefore, we invite you to submit a revised version of the manuscript that addresses the points raised during the review process.

We look forward to receiving your revised manuscript.

Kind regards,

Domiziano Tarantino, MD

Academic Editor

PLOS ONE

Journal Requirements:

All authors have completed the Unified Competing Interest form at www.icmje.org/coi_disclosure.pdf and declare: AJ reports personal fees from Freshfields Bruckhaus Derringer, personal fees from Anthera Pharmaceuticals Ltd, outside the submitted work; All other authors declare no conflicts of interest.

We note that one or more of the authors are employed by a commercial company.

“The funder provided support in the form of salaries for authors, but did not have any additional role in the study design, data collection and analysis, decision to publish, or preparation of the manuscript. The specific roles of these authors are articulated in the ‘author contributions’ section.”

4. Thank you for uploading your study's underlying data set. Unfortunately, the repository you have noted in your Data Availability statement does not qualify as an acceptable data repository according to PLOS's standards.

Reviewers' comments:

Reviewer's Responses to Questions

**Comments to the Author**

1. Is the manuscript technically sound, and do the data support the conclusions?

Reviewer #1: Yes

Reviewer #2: Yes

2. Has the statistical analysis been performed appropriately and rigorously? 

Reviewer #1: Yes

Reviewer #2: Yes

3. Have the authors made all data underlying the findings in their manuscript fully available?

Reviewer #1: Yes

Reviewer #2: Yes

4. Is the manuscript presented in an intelligible fashion and written in standard English?

Reviewer #1: Yes

Reviewer #2: Yes

5. Review Comments to the Author

**Reviewer #1: **The manuscript entitled “Association of intra-articular injection and knee arthroscopy prior to primary knee replacement with the timing and outcomes of surgery: Retrospective cohort study using data from the Clinical Practice Research Datalink GOLD database” introduced that this study was designed to investigate the association between the time and outcome of patients with symptomatic knee osteoarthritis who underwent intra-articular corticosteroid injections and arthroscopic knee surgery and the patient's subsequent primary knee replacement surgery. The biggest advantage of this article is the use of a large data set for analysis. Nonetheless, there are still some flaws in this article, so I suggest revisions. Here are some of the possible problems.

1. In the introduction, the following references on treatment methods for osteoarthritis might be helpful to provide more information and benefit the future readersr: e.g., Injectable hydrogel microspheres in cartilage repair, Biomedical Technology 2023, 1, 18-29./ Functional biomaterials for osteoarthritis treatment: From research to application. Smart Medicne 2022, e20220014./ Microenvironment-responsive nanosystems for osteoarthritis therapy. Engineered Regeneration 2024, 5, 92-110./ Emerging microfluidics for the modeling and treatment of arthritis. Engineered Regeneration 2024, 5, 153-169.

2. The article mentions a three-fold increased risk of SSI within 3 months of IASI and knee replacement, but the event rate is low and difficult to interpret. It is suggested that the author further analyze this conclusion and discuss the reasons for the low rate of interpretation events and the influence on the statistical results.

3. “Both interventions were associated with a lower risk of 1-year mortality

following primary knee replacement, for which there is not a biologically plausible

mechanism, suggesting potential for selection bias due to confounding by indication

for patients receiving these interventions.” Please give a detailed discussion on why there is bias and what are the possible reasons?

4. Is it possible to analyze patients who have received both intraarticular steroid injection and arthroscopic knee surgery as a category?

5. Whether it is possible to cite patients who have received both intraarticular steroid injection and arthroscopic knee surgery as a class, the relevant literature is as recent as possible

6. The good thing about this paper is the large amount of data, but it lacks innovation. Similar studies have been done in the past

7. Please explain why the mortality associated results were seen in the patients who had steroid injection and arthroscopy first mentioned in the paper, and what might be the confounding factors?

**Reviewer #2:** The time and results of primary knee replacement were examined by the authors using an observational retrospective study of linked CPRD gold to determine the relationship between intra-articular injection and knee arthroscopy.

The link between knee arthroscopy and intra-articular injections is a topic of great interest in the field of orthopedics.

The results of this observational analysis imply that poorer outcomes are linked to both knee arthroscopy and steroid injections before primary knee replacement.

Some consideration should be added to the discussion.

Studies have shown that knee osteoarthritis can be effectively treated with injections of sodium hyaluronate and platelet-rich plasma (PRP); nevertheless, in certain cases, PRP may be more advantageous than arthroscopic debridement (10.21037/APM-21-2145). Despite their association with an increased risk of postoperative infection (10.7759/cureus.61649),. Moreover, the role of rehabilitation in those cases should be considered (10.3390/app132413208).

The conclusion paragraph in the article should be different and more argumentative than in the abstract.

6. PLOS authors have the option to publish the peer review history of their article (what does this mean?). If published, this will include your full peer review and any attached files.

Reviewer #1: No

Reviewer #2: No

---

## [Author Response · Author response to Decision Letter 0]

13 Sep 2024

Editors comments:

Please ensure that your manuscript meets PLOS ONE's style requirements, including those for file naming. The PLOS ONE style templates can be found at https://journals.plos.org/plosone/s/file?id=wjVg/PLOSOne_formatting_sample_main_body.pdf and https://journals.plos.org/plosone/s/file?id=ba62/PLOSOne_formatting_sample_title_authors_affiliations.pdf

We have ensured that the manuscript meets the journals style requirements.

Details regarding participant consent has already been included within the manuscript under the heading ‘Ethical approval and informed consent’. This is a retrospective study of routinely collected primary care data from the Clinical Practice Research Datalink (CPRD) where data were already fully anonymised before being accessed for this study. CPRD has ethics approval from the Health Research Authority to support research using anonymised patient data. Researchers apply to CPRD for approval to access anonymised data for a research project. https://www.cprd.com/safeguarding-patient-data

The information below has been included within the manuscript and the same statement has been included to the ‘ethics statement’ field of the submission form:

“No information that can identify a patient is ever sent to CPRD from the contributing GP practices. Because a patient can’t be identified from data a GP practice sends to CPRD, the GP practice doesn’t need to seek a patient’s consent to share data with CPRD (https://www.cprd.com/safeguarding-patient-data). CPRD has obtained ethical approval from a National Research Ethics Service Committee (NRES) for all purely observational research using anonymised CPRD data; namely, studies which do not include patient involvement. The study has been approved by ISAC (Independent Scientific Advisory Committee) for MHRA Database Research) (protocol number 17_127R). CPRD data linked to inpatient HES received on 9th May 2017; linked PROMs data on 25th July 2017. All data were fully anonymized before being accessed for this study by the authors. The authors had no access to information that could identify individual participants.”

All authors have completed the Unified Competing Interest form at www.icmje.org/coi_disclosure.pdf and declare: AJ reports personal fees from Freshfields Bruckhaus Derringer, personal fees from Anthera Pharmaceuticals Ltd, outside the submitted work; All other authors declare no conflicts of interest.

We note that one or more of the authors are employed by a commercial company.

None of the authors are employed by a commercial company. Andrew Judge (AJ) has never been employed by a commercial company. AJ is currently a full time member of academic staff at the University of Bristol since September 2017 (previously full time academic staff member at the University of Oxford (May 2009 to September 2017)). During this period of time AJ conducted external consultancy work, that included payment of fees, as a member of the Data Safety and Monitoring Board (DSMB) for Anthera Pharmaceuticals Inc, San Francisco (13 Dec 2012 to 28 June 2016) and as an expert advisor for international law firm Freshfields Bruckhaus Deringer, London (11 Aug 2014 to 14 Feb 2018). 

The competing interest statement has been amended to make it clear that this was external consultancy work.

This is all as previously described and declared in other recent Plos One publications by Andrew Judge e.g. https://journals.plos.org/plosone/article?id=10.1371/journal.pone.0270274

We have not amended the funding statement to declare a commercial affiliation, where as described above, AJ is not funded by a commercial organisation.

We confirm that existing author contributions statements are accurate.

“The funder provided support in the form of salaries for authors, but did not have any additional role in the study design, data collection and analysis, decision to publish, or preparation of the manuscript. The specific roles of these authors are articulated in the ‘author contributions’ section.”

We have amended the funding section to include the statement “The funder provided support in the form of salaries for authors AJ and MW, but did not have any additional role in the study design, data collection and analysis, decision to publish, or preparation of the manuscript. The specific roles of these authors are articulated in the ‘author contributions’ section.”

As described above, this is not a commercial affiliation. It is related to external consultancy work. The competing interest statement has been amended to be clear this is external consultancy.

The statement has been added to the article, but this is not a commercial affiliation, as stated earlier.

This has been clarified in the cover letter.

4. Thank you for uploading your study's underlying data set. Unfortunately, the repository you have noted in your Data Availability statement does not qualify as an acceptable data repository according to PLOS's standards.

This study used anonymised routinely collected primary care electronic data from the Clinical Practice Research Datalink (CPRD). We are not able to share the data used for this study, and access to CPRD data is subject to protocol approval via CPRD’s Research Data Governance (RDG) Process. We have amended the data sharing statement below:

“Electronic health records are, by definition, considered sensitive data in the UK by the Data Protection Act and cannot be shared via public deposition because of information governance restriction in place to protect patient confidentiality. Access to data is available once approval has been obtained through the individual constituent entities controlling access to the data. The primary care data can be requested via application to the Clinical Practice Research Datalink, secondary care data can be requested via application to the Hospital Episode Statistics from the UK Health and Social Care Information Centre, and mortality data are available by application to the UK Office for National Statistics. Alternatively, access to secondary care data and mortality data is available through linkage requested upon application for primary care data to the Clinical Practice Research Datalink. Information regarding linkage is available from: https://www.cprd.com/linked-data (https://www.cprd.com/research-applications).”

The captions for the supporting information files have been updated as requested including in-text citations as per the journal guidelines.

Reviewer #1

1. In the introduction, the following references on treatment methods for osteoarthritis might be helpful to provide more information and benefit the future readers: e.g., Injectable hydrogel microspheres in cartilage repair, Biomedical Technology 2023, 1, 18-29./ Functional biomaterials for osteoarthritis treatment: From research to application. Smart Medicne 2022, e20220014./ Microenvironment-responsive nanosystems for osteoarthritis therapy. Engineered Regeneration 2024, 5, 92-110./ Emerging microfluidics for the modeling and treatment of arthritis. Engineered Regeneration 2024, 5, 153-169.

The introduction has been updated to included further information on alternative injectable treatments in OA (PRP / biomaterials) and relevant guidance and citations included.

2. The article mentions a three-fold increased risk of SSI within 3 months of IASI and knee replacement, but the event rate is low and difficult to interpret. It is suggested that the author further analyze this conclusion and discuss the reasons for the low rate of interpretation events and the influence on the statistical results.

We identified an infection rate of 0.1% and absolute numbers of infections were low overall. As outlined in our limitations, we planned to capture infections in the same knee within 3 months of the intervention, however it is likely that some infections may not have been captured and recorded in the CPRD dataset. This may have underestimated the event rate and lead to type 1 errors. 

The manuscript has been updated to reflect the above in the limitations section of the study. For interpretation of this finding in the discussion section, we highlight that caution is needed due to the low event rate, and that this may be a chance finding.

3. “Both interventions were associated with a lower risk of 1-year mortality

following primary knee replacement, for which there is not a biologically plausible

mechanism, suggesting potential for selection bias due to confounding by indication

for patients receiving these interventions.” Please give a detailed discussion on why there is bias and what are the possible reasons?

We believe that this is due to selection bias through confounding by indication whereby younger age at Primary Knee Replacement is the key factor in the KA group and is accounting for the reduced mortality. 

We do not believe that there is a biological cause for the reduction in mortality seen with prior IASI users and provide possible explanations for the observed association of reduced risks of mortality with IASI and KA are outlined comprehensively further on in our discussion: 

“It is possible the observed reduction in 1-year mortality associated with prior KA is the result of confounding by indication. Patients who had undergone prior KA where on average 5.7 years younger at the time of primary knee replacement than those that did not undergo prior KA (64.6 vs 70.3) (Table 1). Younger age at primary knee replacement has been associated with lower risk of mortality (28). It may also explain the higher revision rates seen in the KA group, younger patients at time of primary knee replacement are associated with increased life time risk of revision (28). Prior IASI was associated with reduced odds of 1-year mortality (OR=0.76 95%CI (0.58 to 1.00)), however much less so than prior KA, this may be due to more balanced demographics seen in prior IASI users and non-users, especially age (table 1). There is further potential for residual confounding that may be responsible for the reduction in OR for mortality in KA and IASI users. There may be important unknown and unmeasured confounders that where not accounted and controlled for in our analysis.”

4. Is it possible to analyse patients who have received both intraarticular steroid injection and arthroscopic knee surgery as a category?

Prior KA Prior IASI 

 No Yes Total

No 21,314 10,205 31,519

Yes 4,040 2,307 6,347

Missing 437 191 628

Total 25,791 12,703 38,494

The above table outlines the number of patients undergoing both IASI and KA which totals 2,307. This is 6% of our total patient population.

Our current research question for the paper, is whether or not a patient receiving prior IASI, or prior KA, is associated with timing and outcomes of surgery. Acknowledging that these are not mutually exclusive groups, and there will be some overlap of patients who have received both prior interventions. We have addressed this point in our discussion. 

Whilst it would be possible to expand the paper and analyse the exposure as (prior IASI only, prior KA only, prior IASI and KA, no prior IASI and KA), numbers who have received both IASI and KA are small, and it would make the paper more complex expanding the tables, figures, and results section to include this further exposure group.

As such, our preference is simply to report the numbers receiving both IASI and KA within the results section, but not to redo all the analyses with this third exposure group. However, if the editors specifically request us to redo all analyses using this additional exposure group, we would be happy to do so.

5. Whether it is possible to cite patients who have received both intraarticular steroid injection and arthroscopic knee surgery as a class, the relevant literature is as recent as possible

To our knowledge there has been no prior research study that has investigated the impact of patients receiving both IASI and KA prior to knee replacement. Whilst investigating this exposure group was not a research aim of our present study, it could be an interesting topic for future research works. 

6. The good thing about this paper is the large amount of data, but it lacks innovation. Similar studies have been done in the past

Whilst similar studies have been done in the past, the literature remains inconclusive. Our study is one of the largest looking into this important topic that has still to be conclusively answered. The topic is very relevant to orthopaedic surgeons, community GPs and of course patients that commonly unde

---

## [Decision Letter · Decision Letter 1]

30 Sep 2024

Association of intra-articular injection and knee arthroscopy prior to primary knee replacement with the timing and outcomes of surgery: Retrospective cohort study using data from the Clinical Practice Research Datalink GOLD database

PONE-D-24-25859R1

Dear Dr. Judge,

We’re pleased to inform you that your manuscript has been judged scientifically suitable for publication and will be formally accepted for publication once it meets all outstanding technical requirements.

Kind regards,

Domiziano Tarantino, MD

Academic Editor

PLOS ONE

Additional Editor Comments (optional):

Reviewers' comments:

Reviewer's Responses to Questions

**Comments to the Author**

1. If the authors have adequately addressed your comments raised in a previous round of review and you feel that this manuscript is now acceptable for publication, you may indicate that here to bypass the “Comments to the Author” section, enter your conflict of interest statement in the “Confidential to Editor” section, and submit your "Accept" recommendation.

Reviewer #1: All comments have been addressed

Reviewer #2: All comments have been addressed

2. Is the manuscript technically sound, and do the data support the conclusions?

Reviewer #1: Yes

Reviewer #2: Yes

3. Has the statistical analysis been performed appropriately and rigorously? 

Reviewer #1: Yes

Reviewer #2: Yes

4. Have the authors made all data underlying the findings in their manuscript fully available?

Reviewer #1: Yes

Reviewer #2: Yes

5. Is the manuscript presented in an intelligible fashion and written in standard English?

Reviewer #1: Yes

Reviewer #2: Yes

6. Review Comments to the Author

Reviewer #1: “Association of intra-articular injection and knee arthroscopy prior to primary knee replacement with the timing and outcomes of surgery: Retrospective cohort study using data from the Clinical Practice Research Datalink GOLD database”

The authors have well addressed my concerns. I recommend the acceptance of the manuscript.

Reviewer #2: Authors addressed all suggestions, the manuscript is more clear and precise.

No more concerns

Thank You

7. PLOS authors have the option to publish the peer review history of their article (what does this mean?). If published, this will include your full peer review and any attached files.

Reviewer #1: No

Reviewer #2: No

---

## [Editor Report · Acceptance letter]

17 Oct 2024

PONE-D-24-25859R1 

PLOS ONE

Dear Dr. Judge, 

I'm pleased to inform you that your manuscript has been deemed suitable for publication in PLOS ONE. Congratulations! Your manuscript is now being handed over to our production team.

Kind regards, 

on behalf of

Dr. Domiziano Tarantino 

Academic Editor

PLOS ONE